# Axenic Culture of *Caenorhabditis elegans* Alters Lysosomal/Proteasomal Balance and Increases Neuropeptide Expression

**DOI:** 10.3390/ijms231911517

**Published:** 2022-09-29

**Authors:** Huaihan Cai, Ping Wu, Lieselot Vandemeulebroucke, Ineke Dhondt, Madina Rasulova, Andy Vierstraete, Bart P. Braeckman

**Affiliations:** 1Laboratory of Aging Physiology and Molecular Evolution, Department of Biology, Ghent University, 9000 Ghent, Belgium; 2Overseas Pharmaceuticals, Ltd., Room 201, Building C1, No. 11 Kaiyuan Avenue, Huangpu District, Guangzhou 510530, China

**Keywords:** axenic dietary restriction, transcriptomics, lifespan extension, *C. elegans*

## Abstract

Axenically cultured *C. elegans* show many characteristic traits of worms subjected to dietary restriction, such as slowed development, reduced fertility, and increased stress resistance. Hence, the term axenic dietary restriction (ADR) is often applied. ADR dramatically extends the worm lifespan compared to other DR regimens such as bacterial dilution. However, the underlying molecular mechanisms still remain unclear. The primary goal of this study is to comprehensively investigate transcriptional alterations that occur when worms are subjected to ADR and to estimate the molecular and physiological changes that may underlie ADR-induced longevity. One of the most enriched clusters of up-regulated genes under ADR conditions is linked to lysosomal activity, while proteasomal genes are significantly down-regulated. The up-regulation of genes specifically involved in amino acid metabolism is likely a response to the high peptide levels found in axenic culture medium. Genes related to the integrity and function of muscles and the extracellular matrix are also up-regulated. Consistent down-regulation of genes involved in DNA replication and repair may reflect the reduced fertility phenotype of ADR worms. Neuropeptide genes are found to be largely up-regulated, suggesting a possible involvement of neuroendocrinal signaling in ADR-induced longevity. In conclusion, axenically cultured worms seem to rely on increased amino acid catabolism, relocate protein breakdown from the cytosol to the lysosomes, and do not invest in DNA maintenance but rather retain muscle integrity and the extracellular matrix. All these changes may be coordinated by peptidergic signaling.

## 1. Introduction

The potential of the nematode genus *Caenorhabditis* as a powerful model to study the interaction between diet and genetics has long been recognized [1]. While the beneficial effect of dietary restriction on organismal health and aging in *C. elegans* is well appreciated, relatively little is known about the underlying molecular mechanisms, but the field is gradually progressing [2]. *C. elegans* has been used to model the relationship between micronutrients, physiology and metabolism, because of its simplicity and genetic tractability [3,4]. For example, worms fed with *Comamonas* DA1877 bacteria show an altered transcriptional profile compared to worms fed the standard food *E. coli* OP50, and these changes are accompanied by altered developmental rate, fertility, and longevity [5]. Altogether, these findings point out a tight connection between diet, transcriptional responses and resulting life-history traits.

In addition to the standard bacterial food, *C. elegans* can also be cultured axenically, i.e., in chemically defined or semi-defined media without bacteria [6,7]. This allows for direct assessment of the effects of specific nutrients on *C. elegans*. However, postembryonic developmental rate and fertility are compromised in axenic medium, which implies that metabolically active bacteria support optimal worm growth and reproduction [7]. Axenically cultured worms share many traits with worms subjected to dietary restriction (DR), such as slowed development, reduced fertility, a slender appearance, and prolonged lifespan. Hence, the term axenic dietary restriction (ADR) is often applied [8,9]. Like other forms of DR, the effect of ADR on worm lifespan was shown to be largely independent of feeding conditions during development [7,10]. Although lifespan is doubled under ADR, the underlying molecular mechanisms supporting this robust longevity effect still remain enigmatic. As was shown in a genetic screen [9], this mechanism is probably distinct from that of most other DR regimens. In addition, our previous findings showed that worms cultured in axenic medium display increased stress resistance and enhanced mitochondrial function [11,12]. These observations demonstrate that animals under ADR seem healthy but are physiologically different from animals grown under standard bacterial conditions. These physiological differences may support the strong longevity phenotype observed in ADR worms.

To discover which global physiological changes are induced by ADR, we performed RNAseq analysis of worms grown on fully fed (FF) conditions (standard *E. coli* cultures as food source) and worms grown in undefined axenic medium (ADR). We found that ADR up-regulates genes associated with lysosomal activity, amino acid metabolism, muscle integrity, extracellular matrix (ECM), transcriptional regulation, neuropeptides, and small heat stress proteins, while ADR down-regulated genes are related to the proteasome, cell cycle, and DNA replication and repair. These findings provide a handle for future research targeted to these pathways to advance our understanding of the molecular mechanisms of ADR-induced longevity.

## 2. Results

### 2.1. Transcriptomic Profiling of Fully Fed (FF) and Axenic Dietary Restricted (ADR) Worms

Before starting the transcriptomics experiments, we first confirmed the key phenotypes in worms grown under ADR conditions. When cultured in axenic media from the L1 stage, adult worms show a doubling of lifespan (+102%, *p* < 0.001; Appendix A) as well as a small, slender phenotype (Appendix A), which are both consistent with previous reports [9,12].

To gain insight into the molecular mechanisms underlying ADR-induced longevity, we evaluated global changes in *C. elegans* gene expression by comparing *E. coli* OP50 fed (fully fed, FF) and axenically cultured (ADR) adult worms at the second day of adulthood. Principal component analysis (PCA, Figure 1A) and hierarchical clustering analysis (HCA, Figure 1B) show that replicate samples of FF and ADR treatments clearly cluster together in both analyses, although one ADR sample does not cluster well for one principal component. Overall, this indicates high reproducibility between all replicates per treatment. We identified over 1500 differentially expressed genes (DEGs), of which 943 genes are up-regulated and 671 genes are down-regulated (>2-fold change and FDR < 0.1%) in response to ADR (Figure 1C,D; Appendix A). Interestingly, more genes are up-regulated than down-regulated, especially when considering genes showing a 4-fold change in expression (Figure 1D). To obtain a general functional overview of the dataset, the DEGs were first submitted to the PANTHER online analysis tool and grouped into functional clusters based on their molecular function, biological processes, and cellular components (Figure 1E,F) [13]. Strikingly, genes coding for extracellular components are almost exclusively up-regulated (Figure 1F).

### 2.2. Multiple Biological Processes and Pathways Are Modulated by ADR

We further analyzed the set of DEGs with the DAVID tool for functional clustering and discovering gene ontology associations [15,16]. Subsets of ADR-up-regulated genes are involved in lysosomal activity, amino acid metabolism, cytoskeletal and muscular structure and function, extracellular matrix synthesis (e.g., collagens), neuropeptide signaling, and transcription. In contrast, genes required for the ubiquitin–proteasome system, cell cycle, and DNA replication and repair are significantly repressed (Table 1, Appendix A).

### 2.3. Increased Lysosomal Activity May Link to the Breakdown of Nutrients from Axenic Medium

We identified significant changes in gene expression related to the major intracellular protein degradation machineries: lysosomes and proteasomes. A group of genes encoding lysosomal hydrolases and regulating lysosomal biogenesis are up-regulated in ADR worms (Figure 2A,B), while genes involved in ubiquitin conjugation, elongation, and the major proteasome components are down-regulated (Figure 3).

The autophagic–lysosomal pathway is one of the primary pathways responsible for the degradation and recycling of damaged cellular components, such as truncated or misfolded proteins and impaired organelles [17]. Age-associated decline in autophagic–lysosomal function has been suggested to be the main cause of the progressive loss of proteostasis resulting in aging [18]. Genes covering several aspects of lysosomal activity are induced by ADR, including cysteine-type proteases, aspartyl proteases, trypsin-like proteases, carboxypeptidases, as well as lipases and acid phosphatases (Figure 2A,B). Hence, this points to a general elevation of lysosomal activity in ADR worms for the breakdown of proteins as well as lipids. We analyzed whether this expression pattern translates into increased lysosomal enzyme activity in worm samples of the same batch as those used for RNAseq. The activities of cathepsin L and acid phosphatase are indeed significantly enhanced in the ADR worms (Figure 2C,D). A similar trend is seen for cathepsin B but did not reach significance at the 0.05 level. Increased lysosomal activity may underlie the high demand for degradation of nutrients such as peptides from the axenic medium in case this is taken up by endocytosis. On the other hand, the lysosomal activity may reflect the need for increased autophagic recycling of intracellular constituents because of poor nutrient import under ADR. Indeed, autophagy was shown to be indispensable for several interventions that promote longevity, such as dietary restriction and reduced insulin/IGF-1 signaling [19]. It is thought to remove damaged proteins and slow down aging [20,21]. Consistent with this notion, dietary restriction was found to stimulate autophagy [22]. However, we did not observe enrichment of up-regulated autophagy-related genes in ADR-treated worms. In addition, the lifespan-extending effect of ADR was not abolished by RNAi knockdown or mutation of the autophagy-related genes *bec-1*, *lgg-1*, and *hlh-30* during adulthood (Figure 2F–H). HLH-30 regulates the expression of multiple autophagy-related and lysosomal genes and is required for lifespan extension in several *C. elegans* long-lived mutants. This suggests that autophagy is not required for ADR longevity.

Taken together, we hypothesize that increased lysosomal activity under ADR reflects the elevated capacity for breakdown of proteins and peptides from the axenic medium, taken up by endocytosis. This medium is highly enriched for these compounds and contains relatively low amounts of carbohydrates and almost no fat. After lysosomal digestion, free amino acids may be transported to the cytosol and used for protein synthesis or deaminated and burned in the TCA cycle to meet energetic demands.

### 2.4. Decreased Expression of Genes Coding for the Ubiquitin–Proteasome System

The ubiquitin–proteasome system (UPS) is the most important pathway for the degradation of nuclear and cytosolic proteins, and it plays an important role in the regulation of a variety of physiological and pathophysiological processes including tumorigenesis, inflammation, and cell death [23,24]. Proteasome activity decreases with age, which may be linked to the gradual accumulation of abnormal and oxidized proteins in old animals [25]. This suggests that the UPS may be a major regulator of aging [26]. Surprisingly, we found consistent down-regulation of genes coding for proteins of the UPS system such as ubiquitin-conjugating enzymes, ligases, and proteasome subunits (Figure 3). The ubiquitin-conjugating enzyme UBC-18 is known to interact with WWP-1, which mediates dietary-restriction-induced longevity [26,27]. However, RNAi knockdown of *ubc-18* does not abolish ADR longevity (*p* = 0.2991), while *wwp-1*(*ok1102*) mutants reduce it only to some degree in older worms (*p* = 0.0001) (Figure 3C,D). Therefore, UPS is likely dispensable for lifespan extension under ADR.

The decreased expression of proteasomal subunits in long-lived ADR worms seems to conflict with earlier reports where increased proteasomal expression (and thus enhanced clearance of damaged proteins) is linked to lifespan extension. It was found that ectopic expression of the proteasome regulatory particle *rpn-6* is sufficient to confer proteotoxic stress resistance and extends the lifespan in *C. elegans* [28]. The regulation of *skn-1* is tied to the protein degradation machinery and inhibition of proteasomal activity can induce activation of SKN-1 to maintain cellular stress defense [29]. RNAi-targeted inhibition of most proteasome subunits in *C. elegans* causes nuclear localization of SKN-1 and, in some cases, induces transcription of the glutathione-S-transferase *gst-4* [30]. In addition, SKN-1, ELT-2, and the autophagy–lysosome pathway can be activated upon proteasomal inhibition, resulting in increased longevity and enhanced resistance to multiple threats to the proteome, including heat, oxidative stress, and the presence of aggregation-prone proteins [31]. Therefore, inhibition of proteasomal activity seems to activate compensatory processes such as autophagy and stress-response pathways. However, it is unlikely that this feedback loop is of play in ADR worms, as we have shown earlier that SKN-1 is not involved in ADR longevity [9].

### 2.5. ADR Causes an Overall Increase in Amino Acid Metabolism

To maintain homeostasis, a cell must be able to sense its own energy state, assess nutrient availability, and modulate metabolic pathways in a coordinated fashion. Different diets provide nutrients in different proportions and affect the transcription of metabolic genes in different ways. Dramatic changes in gene expression occur following dietary shifts in *C. elegans*, *D. melanogaster*, and *M. musculus* [32,33,34]. The TOR kinase is a major amino acid and nutrient sensor, and inhibition of TOR signaling extends longevity in multiple model organisms [35,36,37].

The undefined axenic medium we use here contains 3% soy peptone, 3% yeast extract, and 0.05% hemoglobin; hence, it is rich in proteins and peptides, less so in carbohydrates, and the level of fat is below the limit of detection [7]. In line with this, we detected an enrichment of up-regulated genes in amino acid metabolism, particularly glutamate metabolism (Figure 4A). In the long-lived *C. elegans daf-2* mutant amino acid catabolism may replenish the TCA cycle [38]. In accordance, in axenically cultured worms, we found an increased expression of the glutamate-hydrolyzing enzyme *asns-2* and glutamate-ammonia ligases *gln-1* and *gln-3*, essential for catalyzing the amination of glutamate into glutamine (Figure 4B). We also found a mild increase (2.3-fold) in the putative glutamate synthase *W07E11.1*, which can interconvert glutamate to glutamine and alpha-ketoglutarate (Figure 4A,B). This expression pattern hints at increased amino acid catabolism in ADR worms. In agreement with this is the clear 7.7-fold increase in expression of the D-amino acid oxidase gene, *daao-1* (Figure 4A), of which the protein product catalyzes the oxidative deamination of D-amino acids with oxygen and this reaction produces hydrogen peroxide, ammonia, and the corresponding 2-oxo acid [39,40]. Other related up-regulated genes include *odc-1*, ornithine decarboxylase, the first enzyme of polyamine biosynthesis, and the cytosolic glutamic-oxaloacetic transaminase (*got-1.2*) connecting amino acids to carbohydrate metabolism (Figure 4B). Despite having a clear signature of up-regulated amino acid metabolism, axenically cultured worms in which such genes are knocked down by RNAi (*asna-2*, *got-1.2*, *gln-1*, *smd-1*, and *adss-1*), do not lose their long-lived phenotype (Appendix A). This suggests that increased amino acid metabolism is not causative to ADR longevity.

### 2.6. Chaperones and Detoxifying Enzymes Are Up-Regulated in ADR Worms

ADR worms show an overall increase in small heat shock proteins, glutathione-S-transferases, and UDP-glucuronosyltransferases (Figure 5). The latter two groups are involved in xenobiotic detoxification [41,42]. In long-lived *daf-2* mutants, small heat shock proteins may be involved in protein stabilization: they may sequester surplus proteins to maintain protein balance [43] and render the *daf-2* proteome less prone to aggregation [44]. Increased protein protection has been suggested to be beneficial for the animal as it prevents protein damage, thereby delaying the gradual collapse of protein homeostasis with age [45]. However, it should be taken into account that heat shock proteins are subject to strong post-transcriptional regulation and mRNA levels may not accurately reflect actual protein levels. Indeed, in the insulin/IGF-1 signaling mutant *daf-2*, transcriptional elevation of HSPs has been described [46,47], while protein levels and turnover rates were shown to be lower than in control strains [48,49].

Taken together, the transcriptional profile of chaperones and the xenobiotic detoxification machinery very much resemble the pattern observed in long-lived *daf-2* mutants. It is unlikely that the up-regulation of these gene sets is, like in *daf-2* mutants, dependent on DAF-16 activity as in ADR worms DAF-16 is not translocated to the nucleus and ADR longevity is independent of DAF-16 [12]. Likely, up-regulation of these protective systems in ADR worms does not support their doubled lifespan but rather increases their thermotolerance and oxidative stress resistance [12].

### 2.7. Axenic Culture Enhances Expression of Genes Involved in Muscle Function

We found 72 DEGs involved in muscle structure and function (Figure 6A–C) and they were almost exclusively up-regulated. This hints to the preservation of structural integrity of the muscles in ADR worms. The up-regulated muscle genes include a list of genes for the heavy and light components of striated muscle and pharyngeal thick filaments, as well as genes encoding for the thin filaments and its interactors (Figure 6A). Additionally, genes required for initial muscle assembly and attachment are also selectively enriched in ADR worms. In addition, transcripts of cytoplasmic proteins of the muscle including those for muscle attachment and calcium signaling and homeostasis are increased in ADR worms (Figure 6C).

We next tested whether these transcriptional changes in muscle-related genes result in visible structural adaptations in axenically cultured worms by using transmission electron microscopy (TEM). Although the absolute surface area of sarcomeres is significantly reduced in ADR worms, the relative proportion of muscular to non-muscular tissue is increased in these animals (Figure 6D–F; Appendix A). This phenotype resembles that of *daf-2* worms described earlier by our group [50] and hints at specific maintenance of muscular tissue in these animals. However, it should be noted that turnover rates of muscle-related proteins are, on average, extremely slow, with protein half-lives often exceeding worm lifespan, indicating that many structural muscle proteins are only produced once in a lifetime [48]. Hence, it is expected that muscles are very stable structures in the worm and maintenance costs of this tissue are relatively low. Taking into account all these observations, the increased presence of muscle-related transcripts in our RNAseq dataset may not be the result of a stimulating regulatory process, but rather a passive consequence of the larger proportion of muscle tissue present in ADR worms.

### 2.8. Increased Expression of Extracellular Matrix Factors and Collagens in ADR Worms

Axenically cultured worms show a clear up-regulation of genes involved in collagen synthesis as well as related processing enzymes such as metalloproteinases (Figure 7A,B). The strongest up-regulated gene (96-fold change) is *col-124*, indicating that its product might be involved in maintenance of the young phenotype of axenically cultured worms. Both *col-14* and *col-183* were specifically switched on in ADR worms and below detection limits in FF worms. COL-14 affects vulva morphology, while COL-183 is a structural constituent of the cuticle and a positive regulator of growth [51]. It is not clear how these properties can be reconciled with the small size of axenic worms (Appendix A). On the other hand, up-regulation of many collagen and collagen-processing genes in ADR worms may be indicative of a thickened cuticle that better protects the worms from their environment and reduces physical damage. However, ultrastructural analysis revealed a reduction in cuticle thickness in ADR worms (Figure 7C). As adult cuticular synthesis mainly takes place before the final molt, the thin cuticle may be the result of poor (axenic) feeding conditions during larval stages. Indeed, when worms are grown on a normal bacterial diet and switched to ADR at adulthood, no difference in cuticle thickness between two-day adult FF and ADR animals is detected (Appendix A). Likely, the functional implication of the collagen up-regulation in ADR worms is more subtle than building a visibly thicker cuticle.

Extracellular matrix proteins, such as collagens, have been linked to *C. elegans* longevity before. Collagen production, or at least a delay in the age-related decline of collagen expression, is a hallmark of several anti-aging interventions and suggests that maintenance of an intact extracellular matrix supports longevity [52]. It seems that ADR is no exception to this rule. Partial reduction in the longevity phenotype of dietary restricted worms or long-lived mutants upon collagen knockdown during adulthood emphasizes the causal relationship between collagen expression and late-life health [53]. However, it should be noted that the sustained production of cuticle collagens in adult worms is also considered a ‘run-on’ phenomenon, resulting in a hypertrophic cuticle at advanced age [54,55].

To investigate the functional importance of collagens in ADR longevity, we analyzed survival of worms submitted to RNAi knockdown of ADR-up-regulated collagens during adulthood. None of these collagen knockdowns clearly affected ADR-mediated lifespan extension (Appendix A).

### 2.9. Down-Regulation of DNA Replication, Repair, and Cell Cycle Genes May Link to Reduced Fertility

Axenically cultured worms show the typical DR-like trait of declined fertility [8,12]. In support of this, we detected a group of down-regulated genes involved in the cell cycle, particularly meiosis (Figure 8A,B). For example, the down-regulated *mei-2* is important for oocyte maturation and its mutation results in defective meiosis [56], while *egg-3* is required for cortical rearrangements at the oocyte surface after sperm entry [57]. In addition, other down-regulated gene clusters are histones and genes involved in DNA replication and repair (Figure 9A,B). Since adult worms are essentially post-mitotic and mitosis only occurs in the germline, we deduce that down-regulation of DNA replication and repair may relate to the limited production of oocytes, which we microscopically confirmed (Figure 8C). It has been reported earlier that worms under ADR experience a certain level of nutrient stress and coordinately allocate more energy to somatic maintenance instead of offspring production [7]. Moreover, signals from the attenuated reproductive system may in turn regulate longevity, particularly the stem cells at the distal region of the gonad [58]. Taken together, the decreased fertility phenotype of ADR worms is strongly reflected in the down-regulation of genes involved in DNA replication and repair and cell cycle genes.

### 2.10. Neuroendocrine Signaling May Orchestrate ADR-Mediated Longevity

Intriguingly, a large group of neuropeptide genes are consistently up-regulated in ADR worms (Figure 10). The number of predicted neuropeptides in *C. elegans* is well over one hundred and most of them fall into three families: the insulin-like peptides (INSs); the FMRFamide (Phe-Met-Arg-Phe-NH2)-related peptides, referred to as FLPs; and the neuropeptide-like proteins, or NLPs [59,60]. In *C. elegans*, there are four proprotein convertases, *kpc-1, egl-3, aex-5,* and *bli-4*, responsible for the maturation of these neuropeptides from proproteins [61]. The proprotein convertase *egl-3* is up-regulated 3.5-fold under ADR, which may link to the increased level of proproteins. However, the exact function and potential redundancy of each of these neuropeptides remains unclear. Among the up-regulated peptide genes we found INS-7, which has been reported to be an important regulator for *daf-2* mutant longevity [62]. As we discovered earlier that the transcriptional co-factor CBP-1 functions in GABAergic neurons to mediate ADR-induced longevity, it is tempting to hypothesize that neuroendocrinal signaling via neuropeptides might be essential in this pathway.

### 2.11. Differentially Expressed Transcription Factors (TF) and Cofactors in ADR Worms

Physiological changes in response to a specific diet require the coordinated regulation of networks of multiple signaling pathways and enzymes within metabolic pathways. Several *C. elegans* transcription factors or co-activators have been related to physiological changes and lifespan extension under a variety of dietary restriction regimens. Examples of these are SKN-1 [63], PHA-4 [64], DAF-16 [65], HSF-1 [66], and CBP-1 [67]. However, most of these were shown to be dispensable for ADR-mediated longevity, except for CBP-1 [9,68].

A large number of TFs show up-regulation in worms under axenic conditions (Appendix A). The function of these in ADR-mediated longevity has not been explored yet. A preliminary lifespan screen, in which we knocked down several of the up-regulated transcription factors by RNAi, did not result in major hits. Only a slight reduction in ADR longevity was found upon *nhr-132* and *xbp-1* knockdown; *nhr-132* is predicted to encode for a nuclear hormone receptor with currently unknown function, and *xbp-1* is an X-box binding protein homolog that is involved in the ER unfolded protein response and has been linked to DR-induced longevity before [69]. However, we found earlier that, under ADR conditions, the ER unfolded protein response is not induced and thus likely is not involved in the observed lifespan extension.

### 2.12. No Clear Patterns in DEGs Involved in Protein Translation

Inhibition of overall protein synthesis rate is a well-established manner to increase lifespan of *C. elegans* [70,71,72]. In addition, the well-studied *C. elegans* longevity mutant *daf-2* shows reduced protein turnover rates [48,50,73,74]. Hence, it would be conceivable to find a consistent decrease in expression of genes related to protein synthesis in ADR worms. Although we noticed differential expression of a set of translation-related genes (Figure 11), no clear directionality could be found, as some were up- and others were down-regulated. However, mitochondrial ribosomal genes seemed to be consistently down-regulated in the ADR worms, which may relate to decreased mitochondrial protein synthesis. It is tempting to speculate that this may lead to mito-nuclear protein imbalance that, in turn, triggers the mitochondrial unfolded protein response (UPRmt) and extends lifespan [75]. Although the UPRmt is indeed induced under ADR conditions, it is not responsible for the strong longevity phenotype of axenically grown worms [10].

## 3. Discussion

The lifespan-doubling effect of ADR in *C. elegans* has been established for almost half a century [76], but most genes essential for longevity in other DR regimens seem not to play a major role in ADR [9,77]. This hints at a unique molecular and physiological signature in axenically cultured worms. By using RNAseq we found that, in young adult ADR worms, lysosomal hydrolases are strongly up-regulated, while components of the UPS system are down-regulated. Opposite regulation of both protein degradation systems may be a compensation reaction to maintain protein homeostasis [78,79], although it is currently not clear which system compensates for which change. An active up-regulation of the lysosomal system could be linked to increased autophagy to provide the cell with components that cannot be retrieved at a sufficient rate from the poor diet. However, we show that autophagy genes are, unlike for many other lifespan-extending interventions [80], not required for ADR longevity. The up-regulation of amino acid metabolism in ADR worms suggests that they actively take up the rich mixture of amino acids and peptides from the axenic medium and use it for cellular maintenance or energy production. It is not yet resolved whether, besides classical transmembrane transport of amino acids through the apical membrane of the intestinal cells, pinocytosis and lysosomal digestion are also involved.

The differential expression of muscle and cell-proliferation-related genes is likely linked to the altered proportional volume of the relevant tissues in the ADR worms. Muscles contain many structural components, and their volume plasticity is likely lower than that of other tissues under varying feeding conditions. Hence, under ADR, they represent a larger share of the total worm volume, and genes expressed specifically in these tissues may therefore be labeled as up-regulated, even if active up-regulation does not occur. The same applies for the gonads, which are underdeveloped in ADR worms and lead to a passive under-representation of genes that are specifically active in this tissue, such as DNA replication, mitosis, and meiosis genes.

The collagen and collagen-processing genes are a conspicuous group of up-regulated genes. As the cuticle in ADR worms is not thickened, their function may be more subtle or, alternatively, a compensatory translation efficiency may result in no major changes at the protein level. Intriguingly, up-regulation of extracellular matrix proteins is a common hallmark for several longevity paradigms [52], and ADR seems not to be an exception to this rule. Finally, ADR worms show up-regulated neuropeptide expression, which may represent the translation of a (lack of) food signal from the neurons or other cells to the rest of the body, provoking systemic physiological responses.

A few other studies have focused on the transcriptional output of axenically grown worms before. In a microarray study using CeMM, a defined axenic culture medium, the top up-regulated gene was identical to the one in our study: the lysosomal cysteine protease *cpr-2* [81]. However, contrary to our results, in CeMM, the *daf-16* transcription factor was up-regulated and several of its downstream targets showed differential expression. DAF-16 is known to be activated under conditions of starvation [82], but not in the chemically undefined axenic medium we used in our study [83]. This may indicate that CeMM confers a more severe nutritional restriction to worms than the undefined axenic mixture we used here and that, at least in part, some other molecular mechanisms may be at play in CeMM longevity.

A similar outcome was reported in a recent study comparing the transcriptional profiles of *C. elegans* and *Litoditis marina* L1 larvae after a 2.5 h exposure to CeMM versus standard *E. coli* food [84]. In addition, here, several lysosomal hydrolases, as well as some daf-16 targets, were found up-regulated in *C. elegans* cultured in CeMM. Interestingly, CeMM did not cause up-regulation of lysosomal hydrolases in *L. marina*, which was also found not to be long-lived in this axenic medium. In an extensive study, Celen et al. (2018) compared gene expression of young adult *C. elegans* wild types grown on standard agar plates seeded with *E. coli*, in S-medium supplemented with *E. coli*, and in liquid axenic CeHR medium (similar to CeMM) [85]. In addition, here, some expression patterns were consistent with our data: UPS components, DNA replication machinery, and genes involved in the reproduction were all down-regulated in axenic conditions. CeHR culturing also up-regulated collagens, but similar to our findings, no structural differences were found in the adult cuticles using SEM. However, alae of L3 cuticles of N2 wild types were more pronounced when worms were grown in CeHR. In the wild-type strain AB1, lysosomal genes were up-regulated in CeHR conditions, but this was not the case for N2 wild types. Finally, neuropeptides were also found up-regulated in CeHR-cultured worms, as well as in worms grown in S-medium supplemented with *E. coli*. Hence, it was concluded that neuropeptide up-regulation may result from liquid culturing rather than from the absence of bacteria. However, a DR phenotype may still be at play, as bacterial uptake in liquid medium is less efficient then in a concentrated lawn on a solid agar surface. Overall, our results point out that lysosomal function, composition of the extracellular matrix, and neuropeptide signaling are likely key processes that deserve further attention in the studies on ADR longevity.

## 4. Materials and Methods

### 4.1. C. elegans Strains and RNAi Bacteria

The wild-type (WT) *C. elegans* used was Bristol N2 male stock. The following two mutant strains were used, JIN1375: *hlh-30(tm1978)* IV and RB1178: *wwp-1(ok1102)* I. These strains were provided by the Caenorhabditis Genetics Center (CGC), Minneapolis, MN, USA. dsRNA expressing *E. coli* HT115 strains targeting candidate genes were from the Ahringer RNAi library. As a control, the bacterial strain containing the empty vector L4440 was used.

### 4.2. Worm Maintenance

Worms cultured on standard NGM plates seeded with *E. coli* bacteria are defined as fully fed (FF), while animals cultured in semi-defined axenic medium are referred to as axenically dietary restricted (ADR) [9]. To avoid contamination, 100 μg/mL ampicillin was added to both culture types. The day of L4-to-adult transition is determined as Day 0 of adulthood. For FF and ADR cultures, it took 3 and 6 days on average to reach this stage at 20 °C, respectively. Worm samples were collected and cleaned by gravitational settling followed by three to five washes in S-basal medium.

Briefly, for axenic culture, the basal medium consisted of 3% soy peptone (Sigma-Aldrich, St. Louis, MO, USA) and 3% yeast extract (Becton-Dickinson, Franklin Lake, NJ, USA), final concentrations (f.c.). Since *C. elegans* is not capable of heme synthesis, after autoclaving, the basal medium was supplemented with 0.05% hemoglobin f.c. (bovine; Serve, Heidelberg, Germany) diluted from a 100× stock in 0.1 M KOH (autoclaved for 10 min). Axenic medium is very rich in nutrients and easily becomes contaminated by microbes. Hence, all equipment and preparations should be handled in a laminar flow cabinet to ensure sterility.

### 4.3. Worm Sampling and Library Preparation

For RNAseq analysis, a parental worm culture was kept on standard NGM plates seeded with *E. coli* OP50, after which gravid worms were double bleached. The resulting eggs were distributed into either axenic liquid medium or NGM plates with *E. coli* OP50. In order to block reproduction and maintenance of synchronous worm populations, we applied floxuridine (5′-fluorodeoxyuridine, FUdR). In our previous work, we noticed that full sterility required higher FUdR concentrations in the fully fed plates compared to axenic medium. We assume that *E. coli* may partially metabolize or inactivate FUdR. Hence, 100 µM and 50 µM FUdR were applied in fully fed and axenic conditions, respectively, to obtain a similar biological effect. Specifically, FUdR (100 µM) was added at L4 stage to the NGM plates, and at the second day of adulthood (D2), the worms were collected from the plates, washed at least five times with S buffer, and stored at −80 °C. In axenic cultures, FUdR (50 µM) was administered at L4 stage after which they were moved to axenic liquid culture [7]. At D2, worms were washed and stored at −80 °C. The total mRNA of each sample was extracted, purified, and prepared using the Ion Total RNAseq kit from Ambion (Austin, TX, USA), and sequencing was completed using the Ion Torrent PGM next generation sequencer, which allows RNAseq on ‘318′ chips (1 Gbase/8 million reads capacity per chip).

### 4.4. Sequence Analysis

Sequence analysis was performed with CLC Bio Genomics Workbench version 7.5 (CLC Bio, Aarhus, Denmark). The procedures include quality control of the raw data, pre-processing of raw data (trimming of sequences, removal of rRNA), mapping to the *C. elegans* reference genome WS, experimental setup, and statistical analysis. Sequences smaller than 50 bp were removed. Default parameters were used for reference mapping. The expression value of each gene was expressed as RPKM (reads per kilobase of exon model per million mapped reads) (Mortazavi et al., 2008), with an unspecific match limit of 10 and maximum number of 2 mismatches. The following mapping parameters were used: mismatchcost: 2, lower indel cost: 1, length fraction: 0.5, similarity fraction: 0.8, and no global alignment. Statistical comparison of RPKM values between the control and axenic libraries was carried out using empirical analysis of differentially expressed genes (DEGs) [86] and multiple comparison correction was performed using a false discovery rate (FDR). Genes were considered to be differentially expressed (DE) when the false discovery rate *p*-value was less than 0.001 with a greater-than-or-equal-to two-fold change in expression.

### 4.5. Functional Analysis and Data Visualization

DEGs were first uploaded to the tool ‘Protein ANalysis THrough Evolutionary Relationships (PANTHER) classification system version 9.0’ for gene identification and general functional characterization (Mi et al., 2013). In order to identify enriched biological themes and GO terms, wormbase gene IDs of DEGS were uploaded to the online bioinformatic tool, the Database for Annotation, Visualization, and Integrated Discovery (DAVID) [16], to look for enrichment in functional clustering and annotation. Functional clustering was performed with high stringency. Biological process terms (DAVID ‘BP Level 2′ categories) were considered significantly enriched when the *p*-value was less than 0.05. Pathway enrichment was determined using DAVID by querying the Kyoto Encyclopedia of Genes and Genomes (KEGG).

Heatmap visualization of DEGs was performed with the MultiExperiment Viewer, part of the TM4 microarray software suite (Saeed et al., 2003; Saeed et al., 2006). Euclidean distances between genes were calculated and used as a distance metric in a hierarchical clustering analysis (HCA), together with an average pairwise distance as linkage method. Pavlidis template matching (PTM) [87] was used to generate gene lists confirming to an expression pattern of interest. The PTM algorithm allows a dataset to be searched for genes of which the differential expression matches a user-defined template profile, based on the Pearson correction between the template and the genes in the dataset [87]. Graphs and statistical tests (including tests for normality) were performed with GraphPad Prism version 6.05 for Windows (GraphPad Software, La Jolla, CA, USA).

### 4.6. Lysosomal Activity Assay

Equal amounts of worms from same batches as the RNAseq samples were frozen and stored at −80 °C. All samples were homogenized by bead beating in 50 mM Na/K phosphate buffer, pH 7.8. Bead beating was performed in a Precellys24 homogenizer, for 30 s at 6800 rpm. To release lysosomal enzymes into the homogenate, CHAPS detergent was added at a final concentration of 1% and the homogenate was kept on ice for 15 min, after which the bead beating step was repeated. Insoluble debris was removed by centrifugation at 14,000 rpm for 8 min at 4 °C. A part of the obtained supernatant was subjected to determination of total protein concentration using a bicinchoninic acid (BCA) assay (Pierce Biotechnology, Rockford, IL, USA). The rest of homogenate was used for acid phosphatase and cathepsin enzyme assays.

### 4.7. Acid Phosphatase Assay

The acid phosphatase assay was performed using a SpectraMax microplate reader. Then, 10 μL of sample was mixed with 90 μL of reaction mixture, containing 4-methylumbelliferyl phosphate (4-MUP) at a final concentration of 97.5 μM in 50 mM citrate buffer, pH 4.6. The excitation and emission wavelengths, optimal for unsubstituted 4-MUP substrate, were set at 360 nm and 450 nm, respectively.

### 4.8. Cathepsin Activity Measurement

Cathepsin activity was measured using a Magic RedTM Cathepsin detection kit (Immunochemistry Technologies, LLC, Davis, CA, USA) based on fluorometry. The reaction mixture consisted of 10 μL of homogenate, 190 μL of sodium acetate buffer (50 mM, pH 4.6), and 8 μL of Magic RedTM 26× solution.

### 4.9. Confocal Imaging

The transgenic reporter strain OD95 was used to observe germline morphology. Animals were grown both in FF and ADR conditions as described above. Day 1 adult worms were randomly picked for confocal imaging. Images were taken with a Nikon C2 confocal microscope using 488 nm excitation wavelength with a 520/30 nm band pass filter for green fluorescence and 561 nm excitation wavelength with a 605/55 nm band filter for red fluorescence.

### 4.10. Transmission Electron Microscopy (TEM) Analysis

Preparation of samples was described previously [50]. Briefly, transverse sections of young adults (D2) of the wild-type N2 strain, cultured on NGM plates seeded with *E. coli* OP50 and axenic conditions, were observed. TEM images were analyzed with specific attention to muscle structure and worm cuticle. Electron microscopy was performed using a Jeol JEM 1010 (Jeol, Tokyo, Japan) operating at 60 kV. Images were digitized using a DITABIS system (Pforzheim, Germany).

### 4.11. Survival Assays

Lifespan assays were performed as described previously [9]. ADR was performed in liquid culture while FF conditions were carried out on solid NGM plates seeded with a lawn of *E. coli* OP50. Briefly, gravid adults were subjected to a microbleaching procedure: about ten worms were transferred to a 10 μL drop of sterile distilled water, 10 μL alkaline bleach (12° hypochlorite and 1M NaOH f.c.) was added, and the mixture was left to incubate for maximally ten minutes until all adults were dissolved.

Before RNAi treatment, worms were cultured on NGM agar plates seeded with *E. coli* OP50 bacteria until adulthood. RNAi was carried out following standard bacterial feeding protocols and performed on adult worms for five days before they were transferred to the experimental conditions. For the FF controls, around a hundred worms of each strain were placed on small NGM plates (ten per plate) seeded with *E. coli* OP50. For ADR, about a hundred worms were transferred to small screw-cap tubes (three to five worms per tube) containing 0.3 mL of liquid axenic medium. Progeny production was avoided adding FUdR at 100 μM and 50 μM f.c. for FF and ADR, respectively. Survival was scored at regular time intervals: daily for the FF condition, every other day for the ADR conditions. In solid conditions, worms were considered dead if they did not respond to gentle prodding with a platinum wire. In liquid conditions, worms were scored dead if no movement could be detected, even after gently tapping the tubes. Worms that died of protruding vulva or crawling off the plates were censored. All lifespan assays were conducted at 20 °C and were repeated at least twice independently (pooled data are shown).

Data were analyzed with the online application for survival analysis (OASIS) as described by Yang et al. [88]. In all cases, lifespan data are indicated as mean ± s.e.m. and *p*-values were calculated using the log-rank (Mantel–Cox) method.

## Figures and Tables

**Figure 1 ijms-23-11517-f001:**
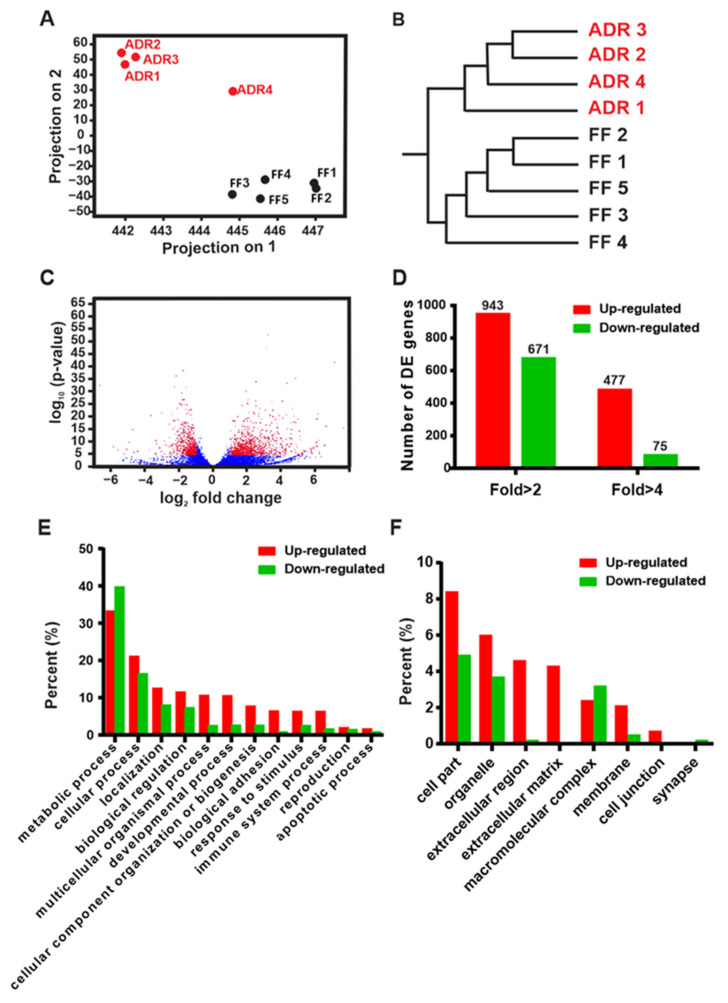
Quality control analysis of the transcriptomic profiles and primary functional clustering of differentially expressed genes by PANTHER online analysis. (**A**,**B**) PCA and HCA analysis of each biological replicate. (**C**) Volcano plot of DEGs (significant hits are marked in red; other detected genes in blue). (**D**) Number of DEGs with two- and four-fold changes in expression. (**E**,**F**) Results of the PANTHER online analysis [13,14] of DEGs submitted: (**E**) biological processes and (**F**) cellular components.

**Figure 2 ijms-23-11517-f002:**
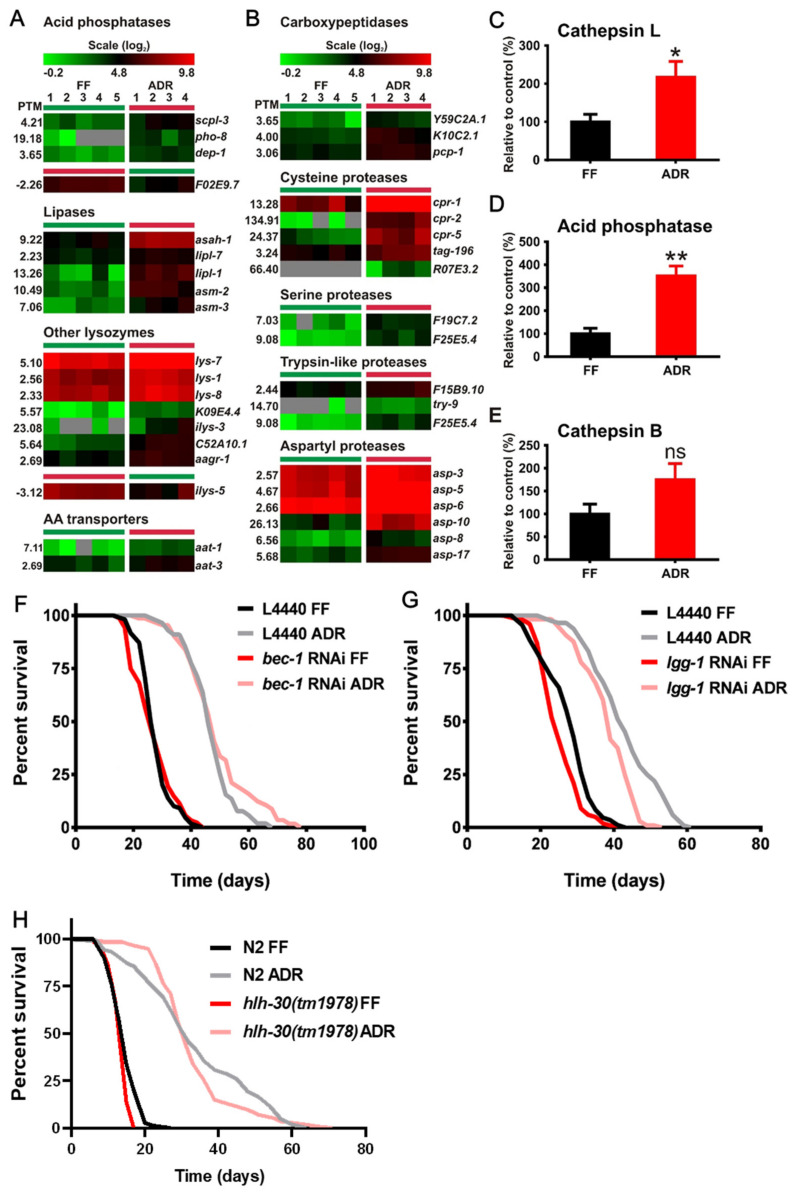
Lysosomal activity is increased in ADR worms. (**A**) Up-regulated expression of genes that encode for acid phosphatases, lipases, other lysozymes, and amino acid (AA) transporters. (**B**) Up-regulated expression of peptidase genes. Cathepsin L (**C**), acid phosphatase (**D**), and cathepsin B (**E**) I activity of FF and ADR worms. Survival analysis of worms with inhibited autophagy: *bec-1* RNAi (**F**), *lgg-1* RNAi (**G**), and *hlh-30*(*tm1978*) (**H**). L4440 = empty RNAi vector, FF = fully fed, ADR = axenic dietary restriction. PTM = Pavlidis template matching, * *p* < 0.05, ** *p* < 0.01, ns: not significant.

**Figure 3 ijms-23-11517-f003:**
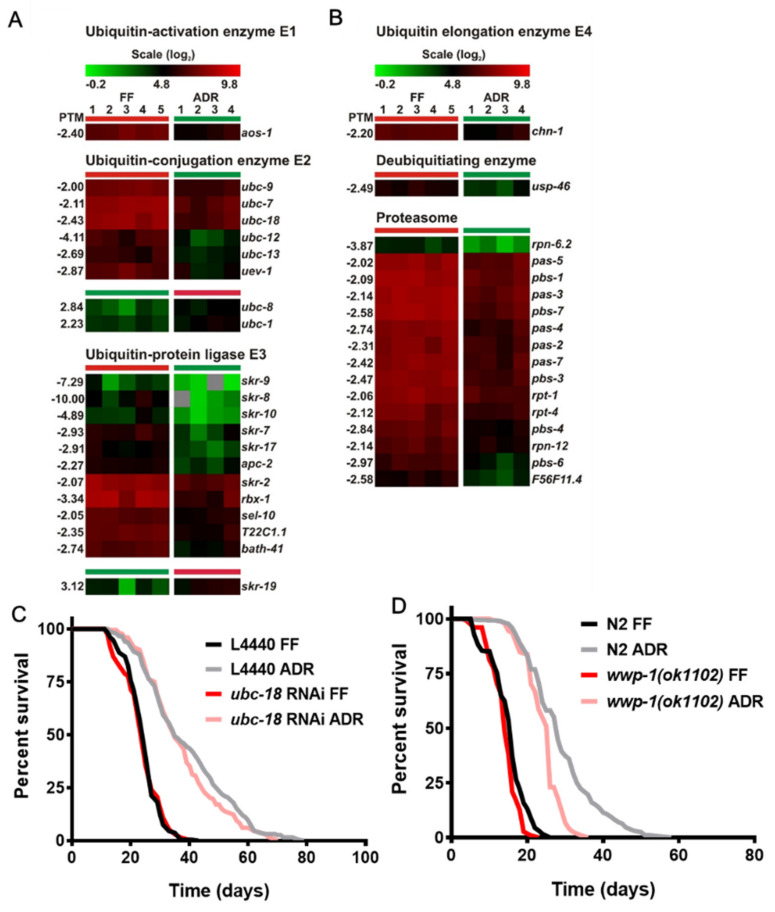
Down-regulated genes of the ubiquitin proteasome system. (**A**) General down-regulation of genes coding for ubiquitin activation enzyme E1, ubiquitin conjugation enzyme E2, and ubiquitin-protein ligase E3. (**B**) General down-regulation of genes in ubiquitin elongation enzyme E4, deubiquitinating enzyme, and proteasomal subunits. Survival analysis of *ubc-18* RNAi (**C**) and *wwp-1* mutants (**D**). L4440 = empty RNAi vector, FF = fully fed, ADR = axenic dietary restriction.

**Figure 4 ijms-23-11517-f004:**
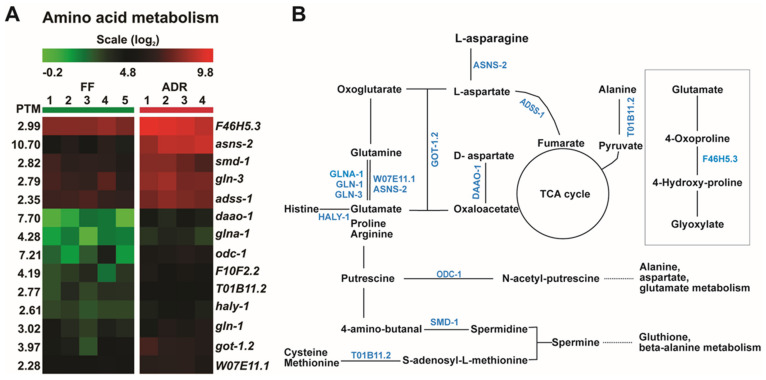
Heatmap and metabolic pathway of DEGs in amino acid metabolism. (**A**) General up-regulation of genes in amino acid metabolism. (**B**) Part of the amino acid metabolic pathway (enzymes are marked in blue). FF = fully fed, ADR = axenic dietary restriction.

**Figure 5 ijms-23-11517-f005:**
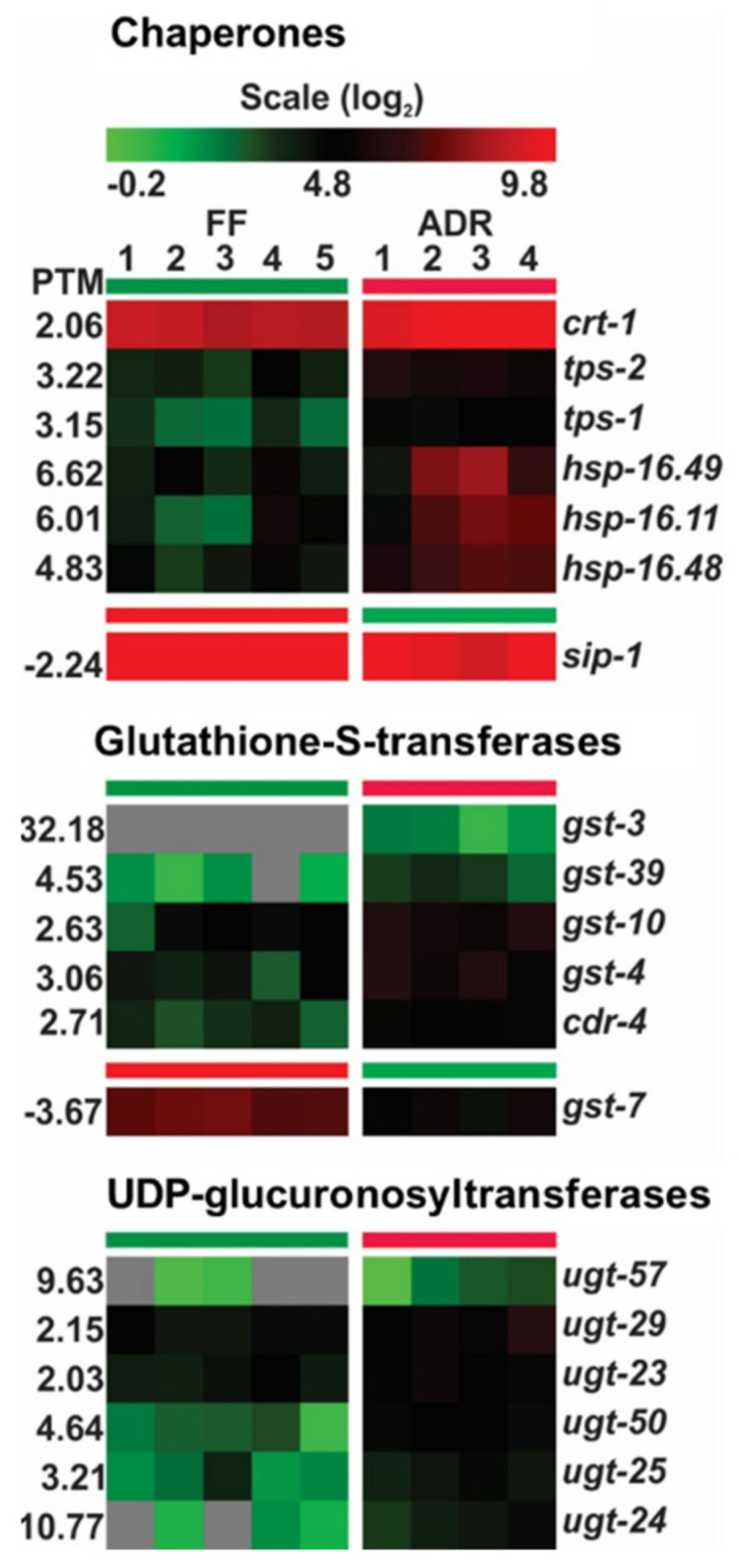
Expression of genes involved in cellular protein homeostasis and xenobiotic detoxification. FF = fully fed, ADR = axenic dietary restriction.

**Figure 6 ijms-23-11517-f006:**
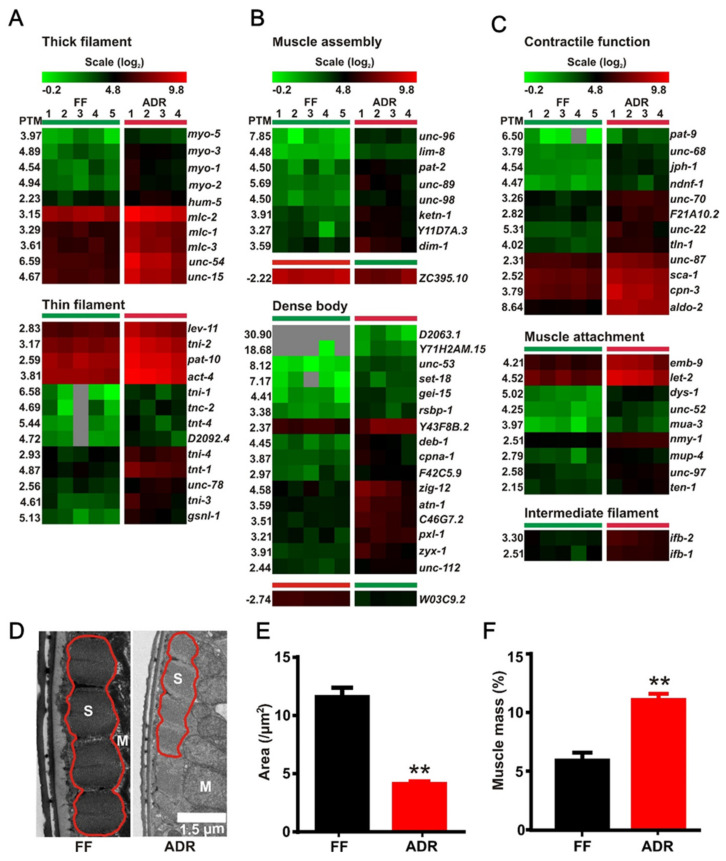
DEGs involved in muscle structure and function. (**A**–**C**) Up-regulated genes of thick filaments, thin filaments, muscle assembly, dense body, and muscle function. (**D**) Transmission electron micrographs (TEM) of muscle structure of both fully fed (FF) and axenically dietary restricted (ADR) worms; S: sarcomere, M: mitochondrion. Sarcomere size (**E**) and relative muscle mass (**F**) in FF and ADR worms, **: *p* < 0.01.

**Figure 7 ijms-23-11517-f007:**
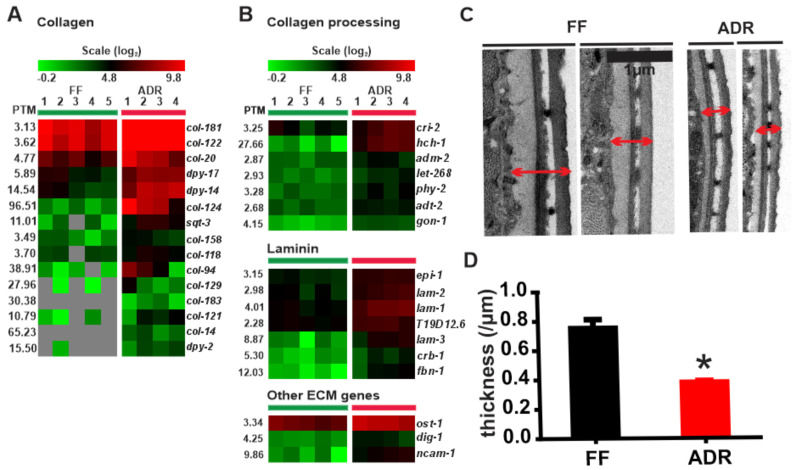
Expression levels of collagens (**A**) and genes involved in collagen processing, laminin, and other extracellular matrix genes (**B**) in fully fed (FF) and axenic dietary restricted (ADR) worms. (**C**) TEM images of cuticles of worms grown in FF (fully fed) and ADR (axenic dietary restriction) conditions. Red arrows = cuticle thickness. (**D**) Cuticle thickness in FF (fully fed) and ADR (axenic dietary restriction) worms, *: *p* < 0.05.

**Figure 8 ijms-23-11517-f008:**
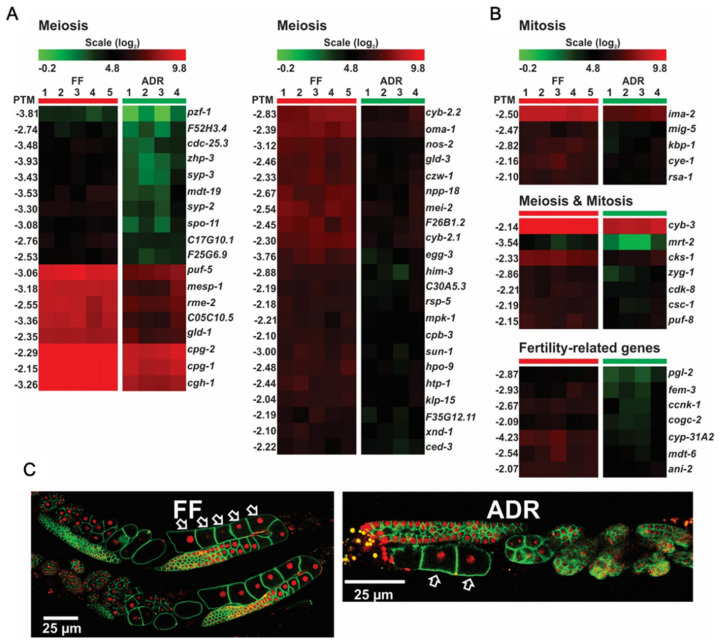
Differential expression of meiosis (**A**) and mitosis (**B**) genes in worms under fully fed (FF) and axenic dietary restricted (ADR) conditions. (**C**) Representative confocal images of gonads of young adults (first day of adulthood) under FF and ADR conditions. Arrows indicate individual oocytes.

**Figure 9 ijms-23-11517-f009:**
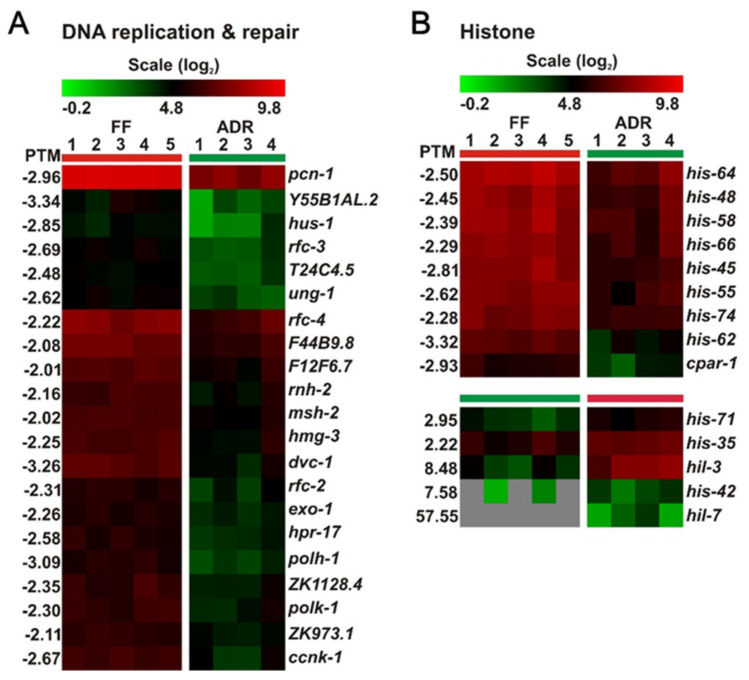
Differential expression of DNA replication and repair genes (**A**) and histones (**B**) in worms under fully fed (FF) and axenic dietary restricted (ADR) conditions.

**Figure 10 ijms-23-11517-f010:**
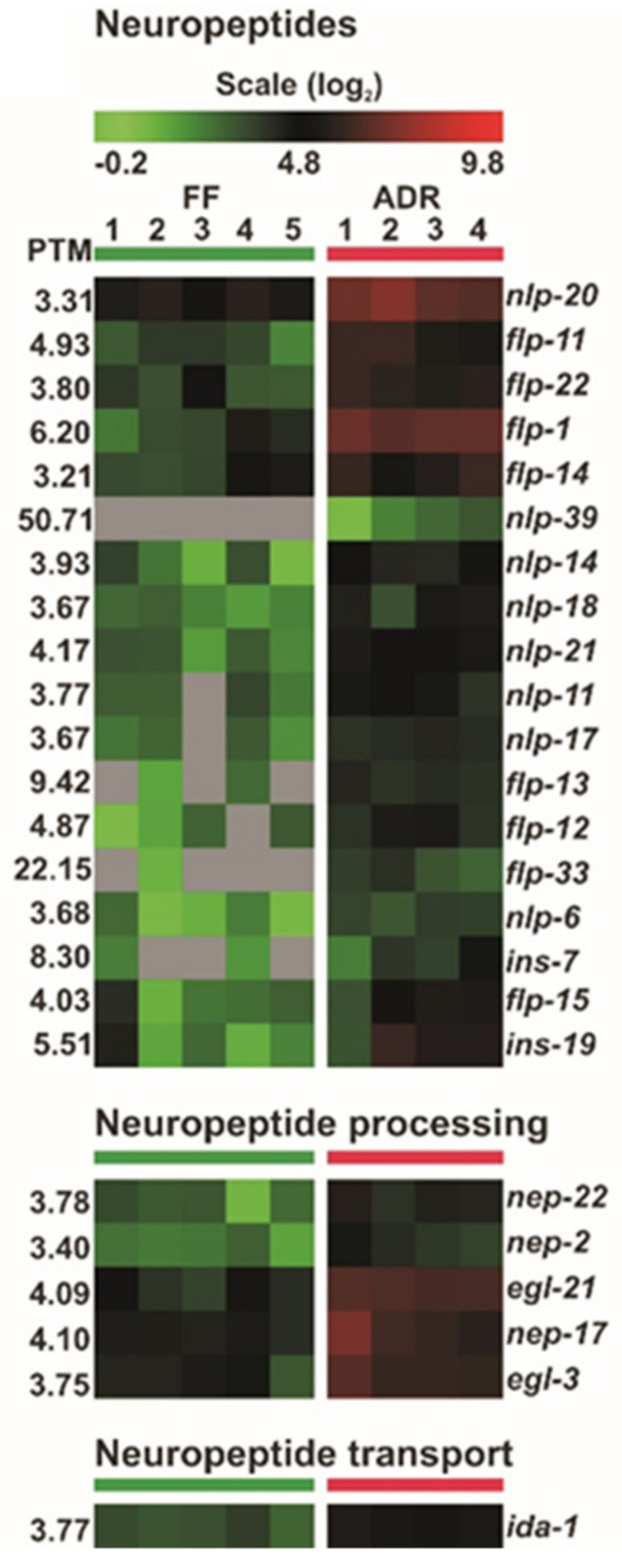
Differential expression of neuropeptide-related genes in worms under fully fed (FF) and axenic dietary restricted (ADR) conditions.

**Figure 11 ijms-23-11517-f011:**
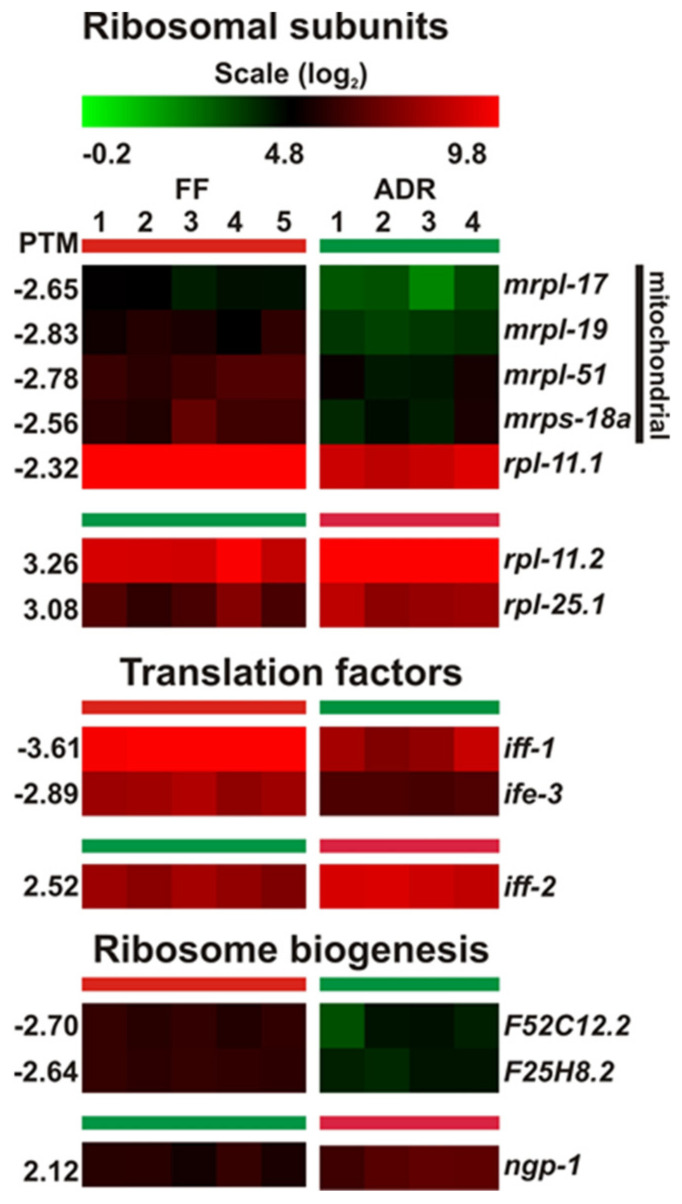
Heatmap of DEGs involved in translation regulation. Panel top: mitochondrial ribosomal subunit genes are generally down-regulated, while the other genes involved in translation and ribosomal biogenesis show up- and down-regulation.

**Table 1 ijms-23-11517-t001:** Enriched pathways of DEGs (from DAVID bioinformatics Resource). Reported *p*-values are the modified Fisher Exact *p*-values or EASE score.

KEGG Pathway Term	Fold	*p*-Value
	Enrichment	
Up-regulated		
Alanine, aspartate, and glutamate metabolism	4.09	5.22 × 10^−3^
Calcium signaling pathway	3.2	9.56 × 10^−3^
Arginine and proline metabolism	3.54	1.09 × 10^−2^
Nitrogen metabolism	4	3.09 × 10^−2^
MAPK signaling pathway	2.32	3.40 × 10^−2^
Down-regulated		
Proteasome	5.22	1.77 × 10^−7^
Mismatch repair	6.03	1.49 × 10^−4^
Ubiquitin mediated proteolysis	2.68	2.86 × 10^−4^
Nucleotide excision repair	3.39	3.38 × 10^−3^
DNA replication	3.29	7.97 × 10^−3^
TGF-beta signaling pathway	3.02	1.30 × 10^−2^

## Data Availability

Data and materials will be made available upon reasonable request.

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
