# Peer review of "Axenic Culture of Caenorhabditis elegans Alters Lysosomal/Proteasomal Balance and Increases Neuropeptide Expression"

_ijms, 2022, doi:10.3390/ijms231911517_

Round 1
Reviewer 1 Report
In this manuscript, C. elegans exhibited altered gene transcription in several pathways under axenic culture. There were up-regulated genes in lysosomal activity, amino acid metabolism, chaperones and detoxifying enzymes, muscle functions, and extracellular matrix factors. There were down-regulated genes in ubiquitin-proteasome pathway, DAN replication/repair, and cell cycle. Over all it is an interesting story to look at the connection between nutrition supply and cellular pathways. However, there are some suggestions I have that are related to the physiological functions in the altered pathways:
1. The physiological functions are carried out by proteins, while mRNA expressions are not necessarily corresponding to protein concentration. It will be more convincing to confirm the key proteins in each altered pathways after the screening.
2. Fig 2, 3, 6, 7, and 8 included functional studies and results, while Fig 4, 5, 9, 10, and 11 lacked related data. Please provide reasons why such studies were missing on some of the pathways.
3. Table1: In the first column, KEGG is the program used to predict, not “enriched pathway”.
4. Figure 2: missing information for L4440 in the figure legend.
Author Response
Dear reviewer,
Please see the attachment.
Regards,
Ping

Reviewer 2 Report
The manuscript represents the effect of axenically cultured C. elegans on the gene expression pattern to understand the response and developement against its culture conditions at molecular level. The comprehensive study of transcriptome analysis in relation to events that occurred in the phenotype is highly commendable. The following problems with the methodology would need to be addressed in future manuscript.
Comments
Line 119: Names need italic. Also, the genes are not italicized through the manuscript. Needs improvement.
Line 492: Fullspelling of FUdR is necessary. Then, why are the concentrations of FUdR different in FF (100 uM) and ADR (50 uM)? Please describe the reason.
Line 494: Were these put in liquid nitrogen, or just putting at -80 C?
Line 512: It would be better to indicate the abbreviation as 'DEG' here.
Line 514: Please describe about data deposition of fastq file for database with accession number.
Line 519: 'gene IDs of DEGs'?
Line 585: The meaning of the FUdR addition should be described at Line 492.
Figure S3: FF and ADR abbreviations should be said at Fig S1.
Author Response
Dear,
Please see the attachment.
Cheers,
Ping
